# Methods of Radiation Wavelength Tuning in Short-Pulsed Fibre Lasers

**Sergey Kobtsev**

Division of Laser Physics and Innovative Technologies, Novosibirsk State University, 630090 Novosibirsk, Russia; s.kobtsev@nsu.ru

**Abstract:** Methods of output wavelength tuning in short-pulsed fibre lasers are analysed. Many of them rely on spectral selection principles long used in other types of lasers. For compatibility with the fibre-optical format, the corresponding elements are sealed in compact, airtight volumes with fibre-optical radiation input and output. A conclusion is presented about the relatively small number of inherently "fibre-optical" ways of tuning the wavelength of radiation. It is demonstrated that the range of output wavelength tuning in short-pulsed fibre lasers may span hundreds of nanometres (even without extension beyond the active medium gain contour through nonlinear effects). From the presented review results, it may be concluded that the search for the optimal tuning method complying with the user-preferred all-PM-fibre short-pulsed laser design is not yet complete.

**Keywords:** fibre laser; mode-lock; short pulse; wavelength tuning

## 1. Introduction

The wavelength tuneability of short-pulsed radiation broadens its application area, and greater tuning ranges result in broader application areas. The wavelength tuning of short-pulsed radiation is complicated by the fact that mode-locking operations must be maintained during the tuning process. Mode locking must be conserved regardless of the changes in conditions and parameters of the generated radiation in the process of tuning [1–3]. Wavelength tuning of short pulses is unusual because the selective element (or effect) should have a broad transmission spectrum that does not restrict the relatively broad spectrum of such pulses. It is desirable that the transmission band of the selective element exceed the width of the short-pulsed spectrum. On the other hand, the working spectral bandwidth of the selective element should fall within the gain contour; otherwise, such an element would not control the radiation spectrum. Therefore, the transmittance (reflectance) bandwidth of the selective element has natural limits on both the narrow and broad sides (Figure 1).

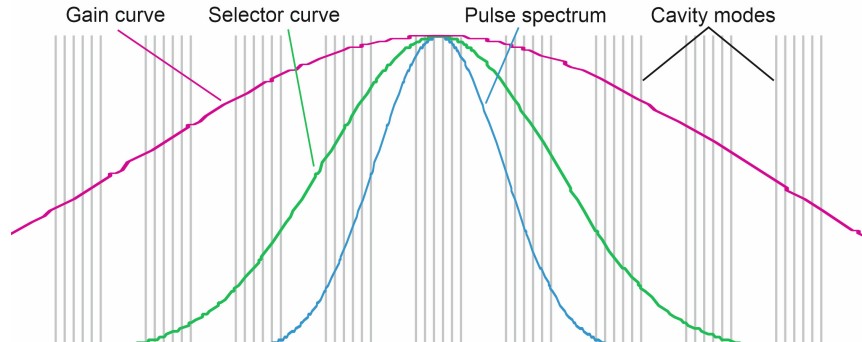

**Figure 1.** Limits imposed on the transmittance (reflectance) bandwidth of a selective element in a short-pulsed laser: $\Delta\lambda_{am} > \Delta\lambda_s > \Delta\lambda_p$, where $\Delta\lambda_{am}$ is the width of the gain contour of the active medium, $\Delta\lambda_s$—the transmittance (reflectance) bandwidth of the selective element; and $\Delta\lambda_p$—the generated radiation pulse spectrum width.

The requirements for the transmission bandwidth of the selective element for short-pulsed radiation are virtually opposite to those imposed on selective elements for continuous-wave radiation (in which case the narrowest possible transmission bandwidth is needed). Sometimes, wavelength tuning is conducted with elements (e.g., polarisation controllers) whose adjustment is badly described, making it almost irreproducible [4–6]. Using such elements for tuning is more suitable for demonstration or research laboratory work and is practically not seen in commercial short-pulsed fibre lasers. We will try to avoid methods of radiation wavelength tuning for CW fibre lasers since this type of tuning does not require the additional condition of maintaining the mode-locked regime. Sometimes, generation at different wavelengths is understood as the production of multi-wavelength radiation or output at certain discrete wavelengths, which is certainly not equivalent to continuous tuning of a single radiation pulse, even though there are cases where simultaneous tuning of several pulses may be implemented [7,8]). Neither are we considering lasers with self-sweeping of their output wavelength (wavelength-swept fibre lasers [9]), since controlling the parameters of such automatic tuning represents a separate task. We aim to identify the methods of radiation wavelength tuning in fibre lasers that produce single short (fs-ns) pulses, are free from the problems just mentioned, and provide stable (or almost unchanged) radiation parameters over the entire tuning range. Since output polarisation is one of those parameters, our focus will be on such laser systems that allow generation with fixed polarisation. In addition, the emphasis will be laid on laser systems implemented in all-fibre format.

One should not forget the classical non-linear approaches to broadening the output spectral range, including harmonic generation [10], parametric generation [11], and soliton self-frequency shift [12]. By now, a substantial range of components and methods have been created for tuning the output wavelength of short-pulsed fibre lasers that need analysis and identification of the best solutions.

This review focuses on practically attained results that consist of reaching the broadest wavelength tuning ranges of short-pulsed lasers with certain pulse parameters. Broad tuning ranges are needed not only for the general improvement of the functional capabilities of lasers (instead of a laser with one fixed output wavelength, the user has access to a universal instrument with a variety of output wavelengths), but also for many applications that require wavelength-tuneable short-pulsed radiation (spectroscopy, multi-photon imaging, medical in vivo measurements, and therapy, among many others). From the viewpoint of pulse duration, the shortest pulses (femtosecond) are also the most in demand because, if needed, they may be relatively easily converted into longer (ps/ns) pulses, whereas the opposite (shortening) is not always possible.

## 2. Elements for Output Wavelength Tuning in Short-Pulsed Fibre Lasers

The elements used for wavelength tuning of the radiation produced by short-pulsed fibre lasers include both those conventionally found in solid-state or other types of lasers (diffraction gratings [13–15], birefringent filters [16–18], prisms [19], acousto-optical filters [20], and so forth) and specifically fibre-optical elements (fibre Bragg gratings [21], specialty fibres [22,23], Sagnac filters [24], and so on) /Figure 1/. It should be noted that certain conventional spectral filters, such as birefringent, can be adapted to all-fibre configurations: plates are replaced by pieces of PM fibre and the role of polarisation-selective Brewster faces is played by polarisation sensitive isolators [18] inserted on either side of the PM fibre. Blazed Bragg gratings [25,26] may also be used as polarisation-sensitive elements. An acousto-optical modulator combined with a tapered fibre [27] or with a tapered fibre and a micro-sphere [28], or with a Bragg grating [29], or the modulator itself as a frequency-shifting element [30] may be used as a spectrally selective element. The acousto-optical effect is used in different ways to implement radiation wavelength tuning [31,32].

Suppression of unwanted radiation frequencies should be stronger in a fibre laser because of its greater gain. Correspondingly, polarising elements in birefringent filters [33] are not tilted Brewster plates but more polarisation-selective elements (polarisation cubes

or polarisation-maintaining fibres, and so forth). Interferometers should have a deeper discrimination curve for reliable suppression of unwanted radiation frequencies [34].

One common disadvantage of most elements used for wavelength tuning in short-pulsed fibre lasers is their incompatibility with the all-fibre concept. Often, tuning elements appear to be compatible with all-fibre configurations, but in their essence, these elements remain volumetric, only placed in a dust cover or an air-tight fibre pig-tailed package (examples of such "fibre" components may be found in [35]). Such implementation of these components introduces significant additional losses, but the usually high gain of fibre lasers still makes such quasi-fibre elements acceptable.

The list of naturally fibre-optical spectrally selective elements (e.g., fibre Bragg gratings) is relatively narrow. Additionally, not all of them allow broad wavelength tuning (spanning the entire spectral gain contour of the active medium or at least a significant portion thereof). Fibre Bragg gratings may be tuned within a short range (several nanometres) by thermal expansion [36]. To broaden the tuning range up to tens of nanometres, mechanical methods are applied (bending, compression, or tension of Bragg gratings) that produce significant variation in the grating period [37,38]. The problem with this method is the possibility of exceeding the threshold of plastic deformation in quartz, above which the modification of the grating period becomes irreversible. Restricted bending of the fibre allows reproducible tuning of the output radiation of mode-locked fibre lasers within a range of ~15 nm [22]. Sometimes, combined (simultaneous thermal and mechanical) action is applied to the fibre with a Bragg grating [39].

Radiation wavelength tuning over an 8 nm range was demonstrated with a multi-mode fibre inserted into the laser cavity [40]. This method is limited to linear cavity configurations and has a relatively narrow wavelength tuning range. The introduction of multi-mode fibre leads to the excitation of higher-order modes in it, whose interference depends upon the spacing between the multi-mode fibre and the cavity reflector. A system including a piece of multi-mode fibre and a cavity mirror behind it may be regarded as a single interference element that can introduce spectrally selective radiation losses.

When transitions between SMF (single-mode fibre) and MMF (multi-mode fibre) are used, other processes may be activated within the cavity (optofluidic, stress optic, thermo-optic, nonlinear), which produce radiation wavelength tuning [41]. It is necessary to mention that the wavelength tuning methods relying on a piece of multi-mode fibre are mostly for proof of concept and not frequently used because of high losses, because it is necessary to use liquid, because mechanically moving parts are required, and so forth.

A fibre-optical Mach–Zehnder interferometer [2] may also be used as a spectrally selective filter. This method's drawback is that one of the interferometer arms must contain an optical variable delay line, a component that contains volumetric elements and introduces high losses. Another interferometer that was used before for radiation wavelength tuning is the Michelson interferometer [42]. Bending a bi-conical tapered fibre in one of the interferometer arms could lead to tuning the laser's output wavelength within a few nanometres.

Radiation wavelength tuning within a range of around 16 nm was demonstrated when using a birefringent Sagnac filter [43] as a spectrally selective element. Tuning was performed at an average radiation power of ~5 mW and a pulse duration of ~700 fs. The Sagnac effect is also used for broad-range (~26 nm) wavelength tuning of sub-picosecond pulses by temperature adjustment of a fibre optical loop mirror [44].

Even broader radiation wavelength tuning (136 nm) was demonstrated in a Tm-doped mode-locked all-fibre laser around 2 μm [45]. Mode locking of the laser's radiation was performed by the effect of non-linear polarisation evolution [46], and wavelength tuning was carried out by irreproducible manipulations with a polarisation controller. The drawbacks of this method not only include the absence of a comprehensible tuning algorithm but also the drift of the polarisation controller settings over time because of the plastic deformation of its fibre.

Extra-cavity wavelength tuning methods also use the mechanism of soliton self-frequency shift in specialty non-linear fibres (for example, suspended-core micro-structured $TeO_2$-$WO_3$-$La_2O_3$ glass fibre) [47,48]. The generation of radiation at a given wavelength is governed by a combination of several parameters, including the wavelengths of the initial radiation and that of the fibre zero dispersion, the fibre length, and the power of the pumping pulses. If only the latter two parameters are varied, the range of continuous wavelength tuning may stretch over several hundred nanometres with substantial (by an order of magnitude and more) variation of the energy carried by the output pulses.

## 3. Electrically Controlled Methods of Radiation Wavelength Tuning

Electrically controllable radiation wavelength tuning of short pulses is desirable in many spectral measurements. In this section, we will discuss the implementation of electrically controlled tuning and tuning methods that may be electrically controllable while retaining the all-fibre laser cavity design.

It should be noted that many of the radiation wavelength tuning methods mentioned heretofore may be electrically controlled if the selective elements (diffraction grating, prism, birefringent filter, etc.) are driven electro-mechanically (by galvanometers, step motors, MEMS, and so on). Nonetheless, these volumetric discrete elements do not, as a rule, conform to the all-fibre design, and therefore they rarely find application outside the laboratory. In addition, the methods by which the selective element is physically moved (by tilting, turning, etc.) suffer from this very fact, suffering from limited speed, accuracy, and life. Configurations with stationary selective elements are possible [49] and may even be electrically driven (through an acousto-optical element), but they do not conform to the all-fibre design.

With the advent of acousto-optical tuneable filters implemented as fibre elements, it became possible to tune the output wavelength of mode-locked all-fibre lasers in wide range electrically [50]. For example, an acousto-optical selector was used to obtain a tuneable generation of sub-picosecond pulses within the range of 100 nm (1705–1805 nm) [51]. Acousto-optical spectral filters that do not conform to the all-fibre format were also used for wavelength tuning of fibre laser pulses within tens of nanometres [52]. Narrow-range (several nanometres) wavelength tuning is possible in certain fibre laser configurations by adjustment of pump radiation power [53]. Tuning of the pulse wavelength is possible by electronically controlled heating of the reflective long-period Bragg grating [54]. An approximately ~15 nm tuning range was demonstrated (when heated up to 500 °C) around 1.5 μm. Electrical control of the wavelength of short pulses may also be possible with the help of diverse exotic devices (for instance, liquid crystal on silicon [55]), but sometimes this leads to the generation of structured noise-like pulses, which at this time are much less used in applications [56]. Wavelength tuning of short pulses may be performed by adjustment of a modulator frequency in a configuration with active radiation mode locking [57,58] (a variation of wavelength-swept fibre laser with controlled scanning).

The ability of the laser cavity length to 'self-adjust', factoring in dispersion, to the modulation frequency under active mode-locking was studied in [59]. It was established that changes in the modulation frequency lead to a related change in the radiation wavelength that corresponds to that cavity length whose inter-mode frequency is equal to or a multiple of the modulation frequency. It was shown that, as the modulation frequency changed by 200 kHz, it was possible to tune the output wavelength by 100 nm.

The market availability of fibre-optical elements for controlling the wavelength of short-pulsed radiation (electronically and manually driven) enables quick implementation of wavelength tuning in a short-pulsed fibre laser. It is necessary to note that they all rely on well-known principles, and many of them incorporate volumetric components with fibre input and output. Among these one may list: spectrally tuneable element (tuning range 1535–1570 nm) with adjustable transmission bandwidth (the 3-dB level of 3, 5, and 6 nm with the possibility of ordering a specific width) [60], spectrally tuneable filter based on a thin interferometer (1525–1565 nm, other ranges possible, bandwidth (FWHM): 1–18 nm) [61], (1000–1100 nm) [62], and others.

As it was mentioned before, controlling short-pulsed radiation wavelengths requires relatively weak spectral selection. This is why devices for wavelength tuning of short pulses emerged quickly, and their design is simple. In essence, the most difficult part of their design is the efficient fibre-optical input and output of radiation.

**4. Broad Wavelength Tuning Ranges of Mode-Locked Fibre Laser Radiation**

The spectral width of the gain contour in fibre lasers spans tens and hundreds of nanometres. Correspondingly, one may expect the achievement of similar wavelength tuning ranges for short-pulsed radiation. In this section, we will mention several publications reporting comparatively broad tuneability ranges of mode-locked fibre lasers. Within the 1 μm wavelength domain, the broadest tuning of 90 nm (980–1070 nm) was achieved in [63]. Tuning was achieved by tilting a highly reflective mirror installed behind a compressor made of two diffraction gratings. Adoption of open volumetric elements for wavelength tuning of fibre laser pulses cannot be considered an optimal solution. Another solution using open volumetric elements is presented in [64], where a tuning range of ~55 nm was attained (1004.99–1059.43 nm) via tilting diffraction grating. In the all-fibre format, the reported tuning range (53 nm, 1023–1076 nm) was slightly lower [65], but that solution cannot be considered optimal either because of three (!) polarisation controllers inside the cavity with practically irreproducible settings. The optical layout of a mode-locked Yb-doped fibre laser without polarisation controllers (all-PM) with a tuning range of 44 nm (1020–1064 nm) is given in [66], where radiation wavelength tuning was conducted with a commercial spectral filter (II-VI WaveShaper1000A/SP, W/S). We would like to emphasise once again the advantages of a solution based on a commercial polarisation-maintaining fibre-optical filter with electronic control. Such a filter, although not having the broadest working range (1015–1065 nm or 1527–1600 nm), is nonetheless able to provide a fairly broad wavelength tuning range of a mode-locked fibre laser, featuring at the same time electronic control and polarisation-maintaining design.

Bismuth fibres enable pulse generation around 1.3 μm. Radiation wavelength tuning over 70 nm (1300–1370 nm) was demonstrated in an actively mode-locked bismuth fibre laser [67]. For wavelength tuning of output pulses, an unspecified tuneable filter was used.

In the range of 1.5 μm, the greatest tuning range so far achieved is 78 nm (1524–1602 nm) [68]. Tuning was conducted by open volumetric diffraction grating, and the optical cavity included a polarisation controller. A somewhat narrower tuning range (75 nm, 1545–1620 nm) [69] was reported with a combination of two polarisation controllers and a fibre-optical attenuator. Although the laser configuration conforms to the all-fibre design, this tuning method cannot be regarded as optimal since proper adjustment of two polarisation controllers is practically not reproducible. A 62 nm (1532–1594 nm) tuning range was obtained by using two open volumetric diffraction gratings [70]. Another solution provided 56 nm-wide tuning (1505–1561 nm) [71], but it is not optimal either because open volumetric elements were present in the laser cavity. Tuning was performed by a commercial spectral filter (7527: 1550–50 OD4, Alluxa) based on a tilting interference element.

Considerably broader wavelength tuning ranges for the output of short-pulsed lasers have been reported over the wavelength range around 2 μm. In a laser with a combined thulium/holmium active medium, it was possible to achieve wavelength tuning within a 200 nm range (1860–2060 nm) [72]. Tuning was performed with the help of an open volumetric diffraction grating. A 153 nm-wide tuning ability (1828.45–1981.35 nm) was implemented in a Tm-doped all-fibre laser, but this implementation not only relies on polarisation controller adjustment (which is practically impossible to repeat), but also on the varied length of the active medium [73]. A slightly shorter radiation wavelength tuning range (136 nm over 1842–1978 nm) was reported in [45] (this work has already been mentioned above), but the mechanism of mode locking employed there (nonlinear polarisation evolution [74]) was once again related to polarisation controllers whose tuning details were not shared. A tuning range of 121 nm (1862–1983 nm) was attained in a Tm-

doped all-fibre laser without polarisation controllers by a fiberised grating-based tuneable filter [75]. The drawback of this configuration is its relatively long output pulses (~200 ps).

Within the 3 μm range, wavelength tuning of ~6-ps pulses across 110 nm (2710–2820 nm) was attained. [76]. For this range, there are still few commercial components available; therefore, tuning was carried out with an open volumetric blazed grating as the wavelength-selective feed-back.

A record-setting radiation wavelength tuning range of 330 nm (2.97–3.30 μm) was attained in a dysprosium-doped fibre laser [77]. Tuning was performed with an acousto-optical filter, and pumping was carried out with an Er:ZBLAN laser emitting at 2.83 nm.

Analysing the reported results, we must note that commercial models of fibre-optical tuneable spectrally selective filters for the wavelength ranges of 1 μm, 1.5 μm, and 2 μm are now successfully used.

The key parameters of the reviewed lasers are given in Table 1.

**Table 1.** Basic parameters of analysed short-pulsed fibre lasers: Δλ—wavelength tuning range; AM—active medium (YDFL—ytterbium-doped fibre laser; EDFL—erbium-doped fibre laser; EYDFL—erbium:ytterbium-doped fibre laser; BDFL—bismuth-doped fibre laser; TDFL—thulium-doped fibre laser; TDFA—thulium-doped fibre amplifier; THDFL—thulium:holmium-doped fibre laser; EZBLAN—$Er^{3+}$-doped ZBLAN fibre laser; DZBLAN—Dy:ZBLAN fibre laser); Δt—pulse width; PRR—pulse repetition rate; M/S—mechanism of tunability/selector; Out—average output power of radiation.

| Δλ, nm | AM | Δt, ps | PRR, MHz | Out, mW | M/S | REF |
|---|---|---|---|---|---|---|
| **90** (980–1070) | YDFL | 1.6–2 | 30 | 3 | Tilting of the HR mirror | [63] |
| **54.44** (1004.99–1059.43) | YDFL | 2.43 | 27.2 | ~4 | Free-space reflective grating | [64] |
| **53** (1023–1076) | YDFL | ~1 | 5.814 | ~0.01 | Adjusting PCs | [65] |
| **44** (1020–1064) | YDFL | 2–8 | 6.12 | ~0.7 | Programmable optical filter | [66] |
| **90** (1015–1105) | YDFL | 0.616–0.895 | 47.71 | 15.2 | Tilting the resonator mirror | [78] |
| **170** (1034–1104) | YDFL | 0.187–0.192 | 24.7 | 7.61 | Tuning the intra-cavity loss | [79] |
| **70** (1300–1370) | BDFL | 1300 | 1.683 | 7 | Tuneable filter | [67] |
| **78** (1524–1602) | EDFL | 0.5–0.94 | 17.5 | 2.4–3.1 | Free-space reflective grating | [68] |
| **75** (1545–1620) | EYDFL | 0.64–1.3 | ~4.4 | 22–95 | Adjusting PCs | [69] |
| **62** (1532–1594) | EDFL | 0.9–1.4 | 32.8 | 1.6–5.3 | Adjusting the width of a slit | [70] |
| **56** (1505–1561) | EDFL | 0.6–2.6 | 250 | 5–15 | Spectral filter | [71] |
| **152** (1740–1892) | TDFL | 2.76 | 3 | ~180 | Bending-induced filtering effect | [80] |
| **200** (1860–2060) | THDFL | <5 | 18.4 | 0.4–1.5 | Free-space reflective grating | [72] |
| **152.9** (1828.45–1981.35) | TDFL | <1000 | 4.866 | 3.2 | Adjusting PCs | [73] |
| **143** (1867–2010) | TDFL | <0.4 | 247.98 | 35 | Adjusting PC | [81] |
| **136** (1842–1978) | TDFL | several ps | 2.6 | 1–3 | Adjusting PCs | [45] |
| **121** (1862–1983) | TDFL | ~200 | 3.82 | 0.1–6.12 | Fibre tuneable filter | [75] |
| **300** (1733–2033) | TDFL | 0.91–6.43 | 14.83 | 1 | Free-space reflective grating | [82] |
| **113** (1918–2031) | TDFL | 509.7 | <0.2 | 1314 | Adjusting the intra-cavity waveplates | [83] |
| **310** (1980–2290) | TDFA | ~100 | 55.42 | 140.8–968 | Pump power | [84] |
| **339** (2007–2346) | TDFL + TDFA | 0.229 | 96 | 3300 | Pump power | [85] |
| **110** (2710–2820) | EZBLAN | 6.4 | 28.9 | 203 | Blazed grating | [76] |
| **330** (2970–3300) | DZBLAN | ~33 | 44.5 | 140 | Acousto-optic tunable filter | [77] |
| **215** (3397–3612) | $Er^{3+}$:ZrF$_4$ | 53 | 14.5–15.5 | 167 | Acousto-optic tunable filter | [49] |

The mechanisms used for creation of wavelength-dependent losses include polarisation effects (birefringent spectral filters [86] introducing minimal losses for radiation that does not change polarisation when passing through the filter; polarisation controller(s) [45,69,73,81] or intra-cavity waveplates [83], whose function is essentially similar to that of a birefringent filter; intensity-dependent variation of birefringence in the cavity [84,85]; Sagnac loop mirror [87] with spectrally controlled reflection), interference effects (various interferometers: Fabry–Pérot [88], Mach–Zehnder [89], and so on), dispersion effects (prisms [90], diffraction gratings [64,68,76,82], acousto-optical filters [49,77]). The majority of the wavelength tuning mechanisms used in pulsed fibre lasers are known; they were earlier applied to other types of lasers. Among the specific tuning methods typical of pulsed fibre lasers alone, one can list those that use the laser cavity fibre or the fibre of the intra-cavity elements.

It is necessary to point out that Table 1 provides parameters for pulsed fibre lasers, in which significant wavelength tuning was achieved. However, not all of the specified tuning ranges are easy to reproduce. As it was already mentioned earlier, wavelength tuning of a pulsed fibre laser achieved through manipulations with a single or multiple fibre-optical polarisation controllers (which manipulations are usually not adequately described) is practically irreproducible. Such tuning methods as the adjustment of the pump radiation power are inconvenient in that the laser wavelength tuning is accompanied by broad variations of the output power. In other methods of broad-range wavelength tuning, predominantly volumetric spectral filters are used with the addition of fibre-optical input and output of radiation or using free-space beams.

Many researchers are trying to achieve a significant wavelength tuning range for pulsed radiation (at least about the width of the laser's gain profile) while keeping a relatively short pulse duration (ideally within the femtosecond range) that would additionally not change too much as the wavelength of the short-pulsed output of a fibre laser is tuned around. Naturally, it is also desirable to keep the pulse shape constant in this process, and all this would be possible in an all-PM-fibre laser configuration, which provides alignment- and maintenance-free operation and minimises the effect of ambient conditions. However, fulfilment of all these requirements at the same time has not been demonstrated to date in any published work. This may be explained by a couple of key causes. First of all, the active medium gain is subject to significant variation in the wavelength tuning process. Secondly, the parameters (transmittance, spectral width of the transmission band, etc.) of the intra-cavity elements also change. As one can see from Table 1, there is no definite dependence between the tuning range width and, for instance, the pulse duration, even though in many publications, short (femtosecond) pulse duration was achieved at relatively narrow wavelength tuning ranges. There are, furthermore, opposite examples when a relatively broad tuning range is achieved at femtosecond pulse duration [81,85].

## 5. Dependence of the Wavelength Tuning Range on the Output Pulse Duration

The publications cited above focused on getting the broadest possible tuning range, electrical control of the radiation wavelength, and also on various effects used for tuning. However, it is also important to consider the pulse duration achievable in the process and how the tuning range is related to the pulse duration. Let us analyse those works that do not rely on polarisation controllers (assuming here that such controllers do not hold a significant promise in consumer products due to drift and lack or repeatability of their settings).

One of the broadest demonstrated wavelength tuneability ranges of ~100 fs pulses is equal to 1050 nm (1.6–2.65 µm) [76]. Such wide tuning was achieved by combining two partial ranges produced by self-frequency shifting solitons emitted by an Er/Tm fibre laser able to generate at two different wavelengths (1.56 and 2 µm). Although the master oscillator was mode-locked due to non-linear polarisation evolution (in other words, at least one polarisation controller was used in its configuration), the wavelength tuning of the

output pulses was performed in a separate extra-cavity fibre and was not directly related to the polarisation controllers.

Further on, sub-picosecond pulse duration (~400 fs) correlates with a significant wavelength tuning range of 143 nm (1867–2010 nm) [81]. The achieved tuning was performed by adjustment of the polarisation controller, which procedure was not explained in detail. Consequently, this report was essentially just a demonstration. In [68], wavelength tuning of sub-picosecond pulses was reported across the range of 78 nm (1524–1602 nm), and, even though the tuning mechanism was not directly related to the polarisation controller, its presence in the cavity was essential to provide mode locking.

Pulses of several picoseconds in duration could be tuned across 300 nm (1733–2033 nm) [82]. The tuning was achieved via a volumetric diffraction grating, while the pulse duration (and the radiation spectral width) could be adjusted by varying the width of a slit installed in the radiation beam reflected from the grating.

Adjustment of the wavelength of 19 ps pulses within 70 nm (1015–1085 nm) with the aid of an open volumetric diffraction grating was shown in [91]. The Authors claim that despite the use of an open volumetric diffraction grating, the laser demonstrated stable generation due to an all-PM fibre configuration.

The presence of polarisation controllers in laser cavities in Refs [92–120] prevents their more detailed analysis, as mentioned earlier.

### 6. Unconventional Methods of Radiation Wavelength Tuning in Mode-Locked Fibre Lasers

In this Section, we will intrinsically explore fibre-optical tuning methods that do not use the mechanisms presented in Figure 2. It is shown in [121] that non-linear effects in a mode-locked fibre laser with a semiconductor optical amplifier (SOA) as the active medium may induce significant (tens of nanometres) changes in the output wavelength despite the presence of a spectral filter in the cavity. Without a spectral filter, SOA may shift the wavelength of short pulses to shorter wavelengths by up to 20 nm [122].

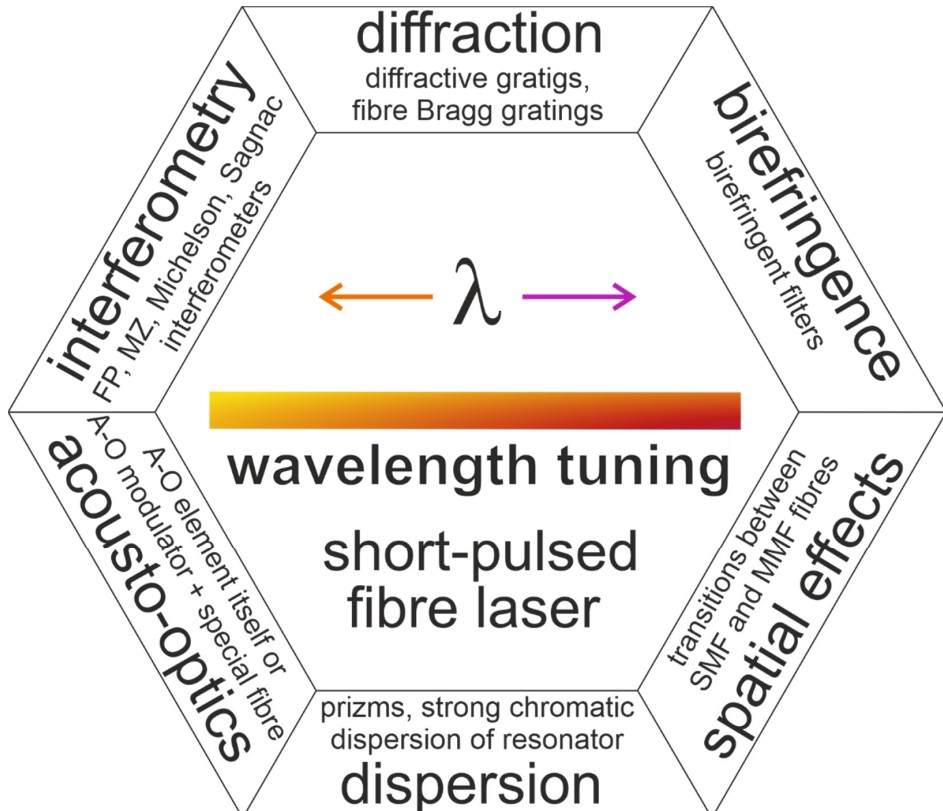

**Figure 2.** Physical effects and respective wavelength selectors of short-pulsed radiation.

Smaller (several nm) wavelength shifts of ~140 fs long optical pulses were observed in a combined (fibre-discrete) cavity with a NALN [123] resulting from pumping radiation power variation [53].

The structure SMF–MMF–SMF was used in [124] as a spectrally selective element for setting the desirable radiation wavelength. Continuous tuning of the transmission spectrum of such a structure was not demonstrated, but setting a fixed radiation wavelength corresponding to the peak transmission of this structure was shown.

Bending a similar structure (SMF-graded-index multimode fibre-SMF) was used to tune the wavelength of sub-picosecond pulses within 23 nm (1040–1063 nm) [125]. Mechanical action on optical fibre could scarcely be considered an optimal solution. Radiation losses in the SMF-MMF transition were ~3 dB.

Taper structure designed as a reflective and, at the same time, spectrally selective element is presented in [126]. Even though this structure allowed wavelength tuning of sub-picosecond pulses over 30 nm (1532–1562 nm), it has the significant demerit of being incompatible with the all-fibre concept.

A saturated absorber interacting with the laser radiation through the evanescent field around a tapered fibre may also function as a polariser [127]. Adjustment of birefringence results in a significant shift of the spectral position of output pulses in a thulium-doped fibre laser (60 nm, 1880–1940 nm).

Laser wavelength tuning with a birefringent filter is well known [128]. However, instead of crystal quartz plates, pieces of PM-fibre are often used in fibre lasers [129,130]. In order to adjust the radiation wavelength, it is necessary to change the length of such a piece (or pieces). This is why selectors lacking dynamic tuneability are seldom used.

Unbalanced Mach–Zehnder interferometers can also be used as spectral filters [131]. The demonstrated wavelength tuning over 35 nm (1527–1562 nm) is not easily reproducible since it was performed with a polarisation controller, and the only description of its adjustment procedure was given as "accurate adjustment".

In a ring-linear laser (where the ring was represented by a NALM), output wavelength tuning was performed by the adjustment of a polarisation controller made of discrete optical elements [132]. Its obvious drawback is its incompatibility with all-fibre cavity configurations.

## 7. Comments

The majority of non-linear fibre-optical methods of radiation wavelength conversion are reviewed in [3], including fibre-optical parametric oscillators, Cherenkov radiation, soliton self-frequency shift, and self-phase modulation-enabled spectral selection. We will not copy this review here but only remind the reader that the review [3] mentions a combination of the highest pulse energy with the wavelength tuning range for different non-linear fibre-optical methods of wavelength conversion. The present review leaves out the methods of extension of the short-pulse generation domain through Raman conversion [133] and super-continuum generation [134]. These topics are well covered in the referenced publications. Another overview of research on broad-range wavelength tuning around the range of 1530–1565 nm as a function of the pulse repetition rate is presented in [61,135].

We will note here that the choice of specifically "fibre-optical" wavelength tuning methods is relatively narrow, and the proposed approaches are not always compatible with the all-fibre design.

Figure 3 shows the wavelength tuning ranges of short-pulsed fibre lasers with an average output power of over 10 mW. Where amplification was used, the tuning range is marked 'A'. It is evident that significant wavelength tuning ranges (110–339 nm) at an average output power exceeding 100 mW (~140–200 mW) were all achieved in a longer-wavelength domain around 2, 3, and 3.5 μm. In more established spectral domains around 1 and 1.5 μm wavelengths, tuning ranges are substantially narrower. It should be once again recalled that Figure 3 presents only short-pulsed radiation tuning ranges at noticeable

average power (>10 mW). Information about lower average radiation powers may be found in Table 1, and the data for continuous-wave radiation can be consulted, for instance, in [14]. It is also desirable to draw the reader's attention to the relative emptiness of Figure 2. The spectral range of ~2.3–2.7 μm is still not spanned by tuning ranges of short-pulsed radiation from fibre lasers delivering significant average powers.

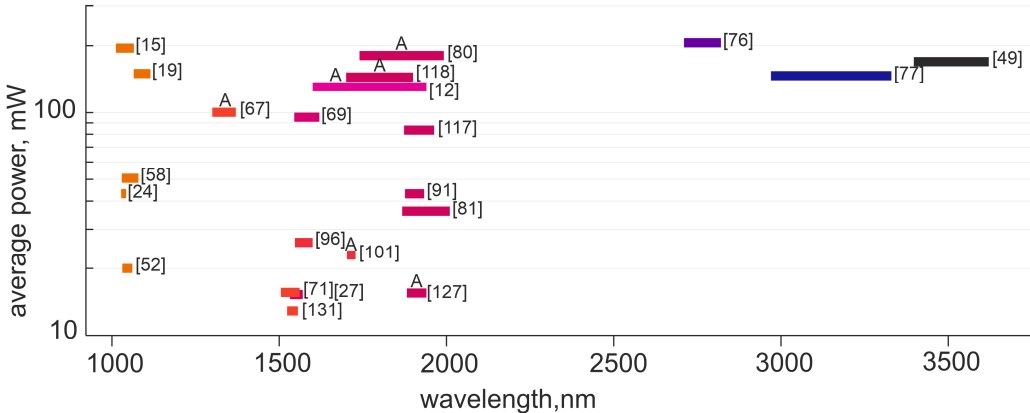

**Figure 3.** Wavelength tuning ranges of short-pulsed fibre lasers with an average output power of over 10 mW [12,15,19,24,27,49,52,58,67,69,71,76,77,80,81,91,96,101,117,118,127,131].

Our analysis demonstrates that there is no significant difference among methods of broad-range wavelength tuning in short-pulsed fibre lasers in various spectral domains. The difference may consist of a dependence on the tuning range width. A relatively narrow tuning range is achievable, for example, by changing the temperature (either of some intra-cavity element or of the ambient air), whereas larger ranges are achieved by traditional selective elements, including free-space reflective gratings or acousto-optic tuneable filters.

Our analysis also identifies no solution that would be optimal for the wavelength tuning of short-pulsed fibre lasers. Existing methods either rely on volumetric (non-fibre-optical) elements or on fibre-optical elements with irreproducible settings (radiation polarisation controllers). The present review stresses the need for the development of new methods of broad-range wavelength tuning tailored specifically to short-pulsed fibre lasers.

## 8. Conclusions

Analysis of wavelength tuning methods for short-pulsed fibre lasers shows that broad-range tuning (by 10–20 nm and more) is possible with commercially available ready-made fibre wavelength filters. These filters use spectrally selective elements that demonstrate good performance as tuning components in other types of lasers. Some of them allow not only broad tuning of the spectral position of their pass band but also its width (in both directions). Such elements may satisfy the needs of most developers and users. However, their working ranges coincide with the working spectral bands of fibre active elements (Yb, Er, and Tm). Therefore, in order to generate short pulses at wavelengths outside the working ranges of well-known active fibres, such mechanisms as Raman or parametric conversion and soliton self-frequency shift are used.

Most methods and means of broad-range wavelength tuning in short-pulsed fibre lasers were adopted from other laser types with some adaptation to the fibre-optical configuration (fibre-optical input/output of radiation). The only method that may be called specific to fibre lasers is swept-tuning, which can provide radiation wavelength scanning across tens of nanometres. This type of tuning, however, is not included in the present overview because dynamic broad-range wavelength tuning is only necessary in a rather narrow application area (OCT, sensor interrogation, etc.).

It should be noted here that many publications use radiation polarisation controllers, either for wavelength tuning of short pulses, for mode-locking, or both. Their adoption is more for demonstration or lab use since their tuning procedure is practically random (which

is why it is not described in many works). Moreover, this tuning has a marked propensity to drift over hours or days. These devices are convenient (simple and inexpensive) for research but are ill-suited for end-user systems.

As it was shown by the analysis performed earlier, the question of the optimal method of wavelength tuning in short-pulsed fibre lasers so far remains open. Many available tuning methods are not compatible with the all-fibre format, whereas keeping that format either reduces that tuning range, produces longer pulses, or results in significant variations of pulse duration (and shape) in the tuning process.

Application in many short-pulsed fibre lasers of wavelength tuning methods successfully tested in other laser types indicates that the same methods may also be optimal for fibre lasers. Inherently fibre-optical methods specific to short-pulsed fibre lasers alone cannot yet compete with them today.

**Funding:** This research was funded by the Ministry of Science and Higher Education of the Russian Federation (FSUS-2020-0036).

**Institutional Review Board Statement:** Not applicable.

**Informed Consent Statement:** Not applicable.

**Data Availability Statement:** All data generated in this study are shown in this manuscript.

**Conflicts of Interest:** The author declares that he has no known competing financial interests or personal relationships that could have appeared to influence the work reported in this paper.

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
