# Peer review of "Methods of Radiation Wavelength Tuning in Short-Pulsed Fibre Lasers"

_photonics, doi:10.3390/photonics11010028_

Round 1
Reviewer 1 Report (New Reviewer)
Comments and Suggestions for Authors
1. This manuscript showed the methods of output wavelength tuning in short-pulsed fiber lasers. While the paper is well organized and well presented, the mechanism of wavelength tuning was not analyzed. The author could summarize the methods based on the tuning mechanism in this manuscript.
2. The authors are suggested to compare and discuss more recent works of mode-locked fiber lasers in the 1-μm range, (i.e. Opt. Lett. 47, 5-8 ,2022; Opt. Lett. 47, 5869-5872, 2022).
3. In the second section, the authors state: " Radiation wavelength tuning over an 8-nm range was demonstrated through insertion into the laser cavity of a multi-mode fiber [40]. This method is limited to linear cavity configurations and has a relatively narrow wavelength tuning range.", but gives no additional information on this process. Could the authors better explain this?
4. In the fifth section, some works of the short-pulsed fiber laser were mentioned, and their wavelength tuning range and the corresponding output pulse duration were shown. However, the issue of how the tuning range is related to the pulse duration was not demonstrated. Some more in-depth discussion should be given in the section.
5. The authors mentioned the high losses of the transitions between SMF (single-mode fiber) and MMF (multi-mode fiber). Please give more details. How much percent is the coupling loss and mode-mismatched loss for different types of fibers? These are quite important information for the readers.
6. The authors mentioned the wide wavelength tuning can be achieved by self-frequency shifting solitons. Could you indicate which parameters of the fiber and input laser pulses are related to the tuning range in this process?
Comments on the Quality of English Language
I have to mention that the language of the manuscript requires some polishing. Certain parts are complicated to follow, and there are numerous mistakes.
Author Response
- This manuscript showed the methods of output wavelength tuning in short-pulsed fiber lasers. While the paper is well organized and well presented, the mechanism of wavelength tuning was not analyzed. The author could summarize the methods based on the tuning mechanism in this manuscript.
Thank you for your appreciation of the structure and presentation of the material. Indeed, the mechanisms of adjustment of the radiation wavelength are not analyzed in this review, as they are described in detail in those works to which references are given. However, in connection with the remark about the lack of analysis of these mechanisms in this work, the following addition is made in the revised version of the article:
The mechanisms used for creation of wavelength-dependent losses include polarisation effects (birefringent spectral filters [129] introducing minimal losses for radiation that does not change polarisation when passing through the filter; polarisation controller(s) [45, 69, 77, 127] or intra-cavity waveplates [102], whose function is essentially similar to that of a birefringent filter; intensity-dependent variation of birefringence in the cavity [89, 107]; Sagnac loop mirror [130] with spectrally controlled reflection), interference effects (various interferometers: Fabry-Pérot [131], Mach-Zehnder [132], and so on), dispersion effects (prisms [133], diffraction gratings [64, 68, 74, 78], acousto-optical filters [49, 75]). The majority of the wavelength tuning mechanisms used in pulsed fibre lasers are known; they were earlier applied in other types of lasers. Among the specific tuning methods typical of pulsed fibre lasers alone, one can list those which use the laser cavity fibre or the fibre of the intra-cavity elements.
- J. Wei, J. Su, H. Lu, K. Peng. A review of progress about birefringent filter design and
application in Ti:sapphire laser. Photonics, v. 10, 1217 (2023). https://doi.org/10.3390/photonics10111217
- R. Li, H. Shi, H. Tian, Y. Li, B. Liu, Y. Song, M. Hu. All-polarization-maintaining dual-wavelength mode-locked fiber laser based on Sagnac loop filter. Opt. Express 26, 28302-28311 (2018). https://doi.org/10.1364/OE.26.028302
- J. Wey, J. Goldhar, G. Burdge. Active harmonic modelocking of an Erbium fiber laser with intracavity Fabry–Perot filters. J. Light. Technol., v. 15, No. 7, 1171-1180 (1997).
https://doi.org/10.1109/50.596963
- F. Yang, L. Zhang, M. Wang, N. Chen, Y. Yao, Z. Tian, C. Bai. Wavelength-tunable mode-locked fiber laser based on an all-fiber Mach–Zehnder interferometer filter. Chin. Opt. Lett. 21, 041401(2023). https://doi.org/10.3788/COL202321.041401
- D. Hanna, R. Percival, I. Perry, R. Smart, Р. Sunit, А. Tropper. An ytterbium-doped monomode fibre laser: broadly tunable operation from 1,010 µm to 1,162 µm and three-level operation at 974 nm. J. Mod. Opt., v. 37, No. 4, 517-525 (1990).
https://doi.org/10.1080/09500349014550601
- The authors are suggested to compare and discuss more recent works of mode-locked fiber lasers in the 1-μm range, (i.e. Opt. Lett. 47, 5-8 ,2022; Opt. Lett. 47, 5869-5872, 2022).
Thank you for the additional references. The review is quite broad, and some works escaped the Author’s attention. The Author is grateful for the provided references, they were included into the revised version of the manuscript.
- In the second section, the authors state: " Radiation wavelength tuning over an 8-nm range was demonstrated through insertion into the laser cavity of a multi-mode fiber [40]. This method is limited to linear cavity configurations and has a relatively narrow wavelength tuning range.", but gives no additional information on this process. Could the authors better explain this?
In relation to this comment, the following was added to the manuscript:
Introduction of multi-mode fibre leads to excitation in it of higher-order modes, whose interference depends upon the spacing between the multi-mode fibre and the cavity reflector. A system including a piece of multi-mode fibre and a cavity mirror behind it may be regarded as a single interference element that can introduce spectrally selective radiation losses.
- In the fifth section, some works of the short-pulsed fiber laser were mentioned, and their wavelength tuning range and the corresponding output pulse duration were shown. However, the issue of how the tuning range is related to the pulse duration was not demonstrated. Some more in-depth discussion should be given in the section.
In response to this comment, the following paragraph was added to the revised version of the manuscript:
Many researchers are trying to achieve a significant wavelength tuning range of pulsed radiation (at least about the width of the laser’s gain profile), while keeping relatively short pulse duration (ideally, within the femtosecond range) that would additionally not change too much as the wavelength of short-pulsed output of a fibre laser is tuned around. Naturally, it is also desirable to keep the pulse shape constant in this process and that all this would be possible in an all-PM-fibre laser configuration, which provides alignment- and maintenance-free operation and minimises the effect of ambient conditions. However, fulfilment of all these requirements at the same time has not been to date demonstrated in any published work. This may be explained by a couple of key causes. First of all, the active medium gain is subject to significant variation in the wavelength tuning process. Secondly, the parameters (transmittance, spectral width of the transmission band, &c) of the intra-cavity elements also change. As one can see from Table 1, there is no definite dependence between the tuning range width and, for instance, the pulse duration, even though in many publications, short (femtosecond) pulse duration was achieved at relatively narrow wavelength tuning ranges. There are, furthermore, opposite examples when relatively broad tuning range is achieved at femtosecond pulse duration [77, 107].
- The authors mentioned the high losses of the transitions between SMF (single-mode fiber) and MMF (multi-mode fiber). Please give more details. How much percent is the coupling loss and mode-mismatched loss for different types of fibers? These are quite important information for the readers.
In the cited work, the radiation losses in the SMF-MMF transition were ~3 dB. This figure was added to the revised manuscript. Other details were not included due to limited volume of the review
- The authors mentioned the wide wavelength tuning can be achieved by self-frequency shifting solitons. Could you indicate which parameters of the fiber and input laser pulses are related to the tuning range in this process?
SFSS was obtained through use of a relatively exotic fibre (suspended-core micro-structured TeO2-WO3-La2O3 glass fibre). This information was added into the revised version of the manuscript.

Reviewer 2 Report (New Reviewer)
Comments and Suggestions for Authors
The author has provided a deep and broad overview of the wavelength tuning of mode-locked fibre lasers. He has selected the most valuable information from a large number of sources dealing with this problem. I think that the manuscript would be useful for the specialists working in the field and can be published in Photonics as it is.
Author Response
The author has provided a deep and broad overview of the wavelength tuning of mode-locked fibre lasers. He has selected the most valuable information from a large number of sources dealing with this problem. I think that the manuscript would be useful for the specialists working in the field and can be published in Photonics as it is.
Thank you for your high opinion of the proposed review. The Author hopes that the presented information and analysis will be of significant utility.

Reviewer 3 Report (New Reviewer)
Comments and Suggestions for Authors
In the paper, Sergey Kobtsev summarized and reviewed the methods of radiation wavelength tuning in short-pulsed fibre lasers. This manuscript accurately pointed out the limits imposed on the transmittance (reflectance) bandwidth and physical effects and respective wavelength selectors of short-pulsed radiation. This review will boost up this field, in my opinion. Congratulations!
The English presentation and language are both clear to me.
Author Response
In the paper, Sergey Kobtsev summarized and reviewed the methods of radiation wavelength tuning in short-pulsed fibre lasers. This manuscript accurately pointed out the limits imposed on the transmittance (reflectance) bandwidth and physical effects and respective wavelength selectors of short-pulsed radiation. This review will boost up this field, in my opinion. Congratulations!
Thank you for your high opinion of the review. The Author is hopeful that this work will stimulate development of more advanced wavelength tuning methods for short-pulsed fibre lasers.

Reviewer 4 Report (New Reviewer)
Comments and Suggestions for Authors
The authors present a review study on the methods of wavelength tuning in short-pulsed fiber lasers with fixed polarization (wavelength swept fiber lasers are not discussed). The emphasis is laid on all-fiber cavity laser systems.
1. Although the authors give an almost complete overview on previous works, they did not elaborate on the physics of wavelength tuning mechanisms. Therefore, I suggest the authors to address this point.
2. I also suggest the authors to improve the abstract especially in terms of language.
3. In section 5, the authors present the dependence of the wavelength tuning range on the output pulse duration. Can you show how far the wavelength tuning range is related to the pulse width? In other words, how far does the pulse width change throughout the tuning range? What about the shape of the output pulse spectrum?
4. Generally, there is a lack of explanation (we have just citations).
5. The conclusion should be supported by new suggestions or open questions.
This work could be of interest to the readers and be granted for publication, only if these points are considered.
Comments on the Quality of English LanguageThe language needs to be improved, particularly in the abstract and conclusion.
Author Response
The authors present a review study on the methods of wavelength tuning in short-pulsed fiber lasers with fixed polarization (wavelength swept fiber lasers are not discussed). The emphasis is laid on all-fiber cavity laser systems.
- Although the authors give an almost complete overview on previous works, they did not elaborate on the physics of wavelength tuning mechanisms. Therefore, I suggest the authors to address this point.
An analysis of physical mechanisms of radiation wavelength tuning (see above) was added to the revised manuscript.
- I also suggest the authors to improve the abstract especially in terms of language.
The Abstract was revised, and language corrected.
- In section 5, the authors present the dependence of the wavelength tuning range on the output pulse duration. Can you show how far the wavelength tuning range is related to the pulse width? In other words, how far does the pulse width change throughout the tuning range? What about the shape of the output pulse spectrum?
As it has already been mentioned in the responses above, there is no definable dependence of the pulse duration upon the tuning range width. A comment to that effect was added to the revised manuscript.
- Generally, there is a lack of explanation (we have just citations).
An explanation of the factors that may affect the pulse duration (and the shape of the output pulse spectrum) was added to the revised manuscript. The added text is given above.
- The conclusion should be supported by new suggestions or open questions.
In response to this comment, the following was added to the Conclusion:
As it was shown by the analysis performed earlier, the question of the optimal method of wavelength tuning in short-pulsed fibre lasers so far remains open. Many available tuning methods are not compatible with the all-fibre format, whereas keeping that format either reduces that tuning range, or produces longer pulses, or results in significant variations of pulse duration (and shape) in the tuning process. An explanation of the factors that may affect the pulse duration (and the shape of the output pulse spectrum) was added to the revised manuscript. The added text is given above.

Reviewer 5 Report (New Reviewer)
Comments and Suggestions for Authors
The review article overviews the existing method of radiation wavelength tuning in short-pulsed all fiber lasers and points out the shortcomings especially when a broad wavelength tuning is desired. Various tuning methods are summarized in four different main categories and a summary table, providing the readers ample and focused information. The outlook on the possible methods that could be employed to achieve such broad wavelength tuning has been discussed in the conclusion section. The article is written very well, which makes it easier to read and comprehend.
Most review articles are very long as they try to cover everything from the basics to all the graphics. In that sense, I would say this article is concise and precise.
However, I still feel like readers would benefit from some representative graphics and governing equations based on which the tuning methods discussed for each category. Also in the introduction section, a survey of application areas that demands such broad range tuning of wavelengths in ns/ps/fs pulsed fiber lasers would be helpful to justify the such technological need and the importance of the review article itself thereof.
Author Response
The review article overviews the existing method of radiation wavelength tuning in short-pulsed all fiber lasers and points out the shortcomings especially when a broad wavelength tuning is desired. Various tuning methods are summarized in four different main categories and a summary table, providing the readers ample and focused information. The outlook on the possible methods that could be employed to achieve such broad wavelength tuning has been discussed in the conclusion section. The article is written very well, which makes it easier to read and comprehend.
Thank you for your favourable opinion of the proposed article.
Most review articles are very long as they try to cover everything from the basics to all the graphics. In that sense, I would say this article is concise and precise.
However, I still feel like readers would benefit from some representative graphics and governing equations based on which the tuning methods discussed for each category. Also in the introduction section, a survey of application areas that demands such broad range tuning of wavelengths in ns/ps/fs pulsed fiber lasers would be helpful to justify the such technological need and the importance of the review article itself thereof.
Thank you for your comment. I would like to note that the proposed review did not aim to provide a fundamental explanation of the wavelength tuning mechanisms in short-pulsed lasers. Such elucidation may be found in numerous monographs, articles, and on-line resources. In this review, we will limit ourselves to a rather short summary of these mechanisms (a corresponding text was added to the revised manuscript and also provided above). In this review, the primary focus is on the practically achieved results, i.e. the widest tuning ranges of short-pulsed fibre lasers with the measured pulse parameters. The proposed review makes it clear that the solutions demonstrated so far are not optimal.
In connection with this comment and the need to specify the application areas of broadly tuneable short-pulsed (within the ns/ps/fs ranges), the following paragraph was added to the revised manuscript:
This review focuses on practically attained results that consist in reaching the broadest wavelength tuning ranges of short-pulsed lasers with certain pulse parameters. Broad tuning ranges are needed not only for general improvement of functional capabilities of lasers (instead of a laser with one fixed output wavelength the user has access to a universal instrument with a variety of output wavelengths), but also for many applications that require wavelength-tuneable short-pulsed radiation (spectroscopy, multi-photon imaging, medical: in-vivo measurements and therapy, many others). From the viewpoint of pulse duration, the shortest pulses (femtosecond) are also the most in demand because if needed, they may be relatively easily converted into longer (ps/ns) pulses, whereas the opposite (shortening) is not always possible.

Round 2
Reviewer 4 Report (New Reviewer)
Comments and Suggestions for Authors
Although the presented review addresses an interesting topic (wavelength tuning in short-pulsed fiber lasers), it is hard to identify any additional contribution to the scientific community.
Although the previous tuning methods have been well summarized, the physics of wavelength tuning mechanisms was not analyzed (we have just citations) and no new ideas or approaches are proposed. Moreover, the issue of how far the pulse width and spectrum change throughout the tuning range was not elaborated.
Therefore, I would not recommend the publication of the manuscript.
Comments on the Quality of English LanguageThe quality of English is okay.
Author Response
Indeed, the mechanisms of adjustment of the radiation wavelength are not analyzed in this review, as they are described in detail in those works to which references are given. However, in connection with the remark about the lack of analysis of these mechanisms in this work, the following addition is made in the revised version of the article:
The mechanisms used for creation of wavelength-dependent losses include polarisation effects (birefringent spectral filters [129] introducing minimal losses for radiation that does not change polarisation when passing through the filter; polarisation controller(s) [45, 69, 77, 127] or intra-cavity waveplates [102], whose function is essentially similar to that of a birefringent filter; intensity-dependent variation of birefringence in the cavity [89, 107]; Sagnac loop mirror [130] with spectrally controlled reflection), interference effects (various interferometers: Fabry-Pérot [131], Mach-Zehnder [132], and so on), dispersion effects (prisms [133], diffraction gratings [64, 68, 74, 78], acousto-optical filters [49, 75]). The majority of the wavelength tuning mechanisms used in pulsed fibre lasers are known; they were earlier applied in other types of lasers. Among the specific tuning methods typical of pulsed fibre lasers alone, one can list those which use the laser cavity fibre or the fibre of the intra-cavity elements.
- J. Wei, J. Su, H. Lu, K. Peng. A review of progress about birefringent filter design and
application in Ti:sapphire laser. Photonics, v. 10, 1217 (2023). https://doi.org/10.3390/photonics10111217
- R. Li, H. Shi, H. Tian, Y. Li, B. Liu, Y. Song, M. Hu. All-polarization-maintaining dual-wavelength mode-locked fiber laser based on Sagnac loop filter. Opt. Express 26, 28302-28311 (2018). https://doi.org/10.1364/OE.26.028302
- J. Wey, J. Goldhar, G. Burdge. Active harmonic modelocking of an Erbium fiber laser with intracavity Fabry–Perot filters. J. Light. Technol., v. 15, No. 7, 1171-1180 (1997).
https://doi.org/10.1109/50.596963
- F. Yang, L. Zhang, M. Wang, N. Chen, Y. Yao, Z. Tian, C. Bai. Wavelength-tunable mode-locked fiber laser based on an all-fiber Mach–Zehnder interferometer filter. Chin. Opt. Lett. 21, 041401(2023). https://doi.org/10.3788/COL202321.041401
- D. Hanna, R. Percival, I. Perry, R. Smart, Р. Sunit, А. Tropper. An ytterbium-doped monomode fibre laser: broadly tunable operation from 1,010 µm to 1,162 µm and three-level operation at 974 nm. J. Mod. Opt., v. 37, No. 4, 517-525 (1990).
https://doi.org/10.1080/09500349014550601
Reviewer 5 Report (New Reviewer)
Comments and Suggestions for Authors
Thank you for responding to reviewers concerns and comments. The article provides good review, analysis and perspective on the methods of wavelength tuning in short pulsed laser fiber lasers which will be valuable in the field.
Author Response
Thank you for your high opinion of the review. The Author is hopeful that this work will stimulate development of more advanced wavelength tuning methods for short-pulsed fibre lasers.
This manuscript is a resubmission of an earlier submission. The following is a list of the peer review reports and author responses from that submission.
Round 1
Reviewer 1 Report
Comments and Suggestions for Authors
Overall this is a good review with a large set of references that will be of interest to readers.
The main drawback, in my opinion, is that it is written rather generally and presents little graphical information. On the example of the analyzed works it is possible to provide schemes of lasers using different methods of wavelength tuning. In addition, a more detailed analysis with a brief indication of some parameters of the reviewed lasers should be added. This would only improve this review and make it more useful for the readers.
As a summary of the review, it is good to conclude about the possibility of using different wavelength tuning methods for lasers of different spectral ranges. Are there any limitations or difficulties in using wavelength tuning methods in different spectral ranges?
There are typos and missing dots at the end of sentences. These should be corrected.
Reviewer 2 Report
Comments and Suggestions for Authors
The authors have done a great job of reviewing the existing methods of wavelength tuning in short-pulsed fibre lasers.
If the authors deem it necessary, they can add tables of tuning parameters for dirrerent lasers or may be some diagrams (similar to Figure 1) illustrating the material in each paragraph to make it easier to understand.
In paragraph "Elements for output wavelength tuning in short-pulsed fibre lasers", string 1-2, the verb "are used" must be in the end of the sentence.
Reviewer 3 Report
Comments and Suggestions for Authors
In this manuscript, the authors demonstrated and analysed the methods of output wavelength tuning in short-pulsed fibre lasers are analysed. Furthermore, there are many approaches on spectral selection principles in other types of lasers introduced. However, chiefly due to lack of novelty and volume of the findings I cannot recommend the manuscript for publication in "Photonics". The following questions should be responded reasonably.
1. There are few figures in the manuscript, which is not acceptable.
2. The authors should provide the tables about the research progress on the wavelength tuning in short-pulsed fibre lasers. Otherwise these statements can come across as abrupt or confusing.
3. There is the statement “In analysis of the reported results, it must be noted that commercial models of fibreoptical tuneable spectrally selective filters for the wavelength ranges of 1 μm and 1.5 μm are being successfully used”. However, now the fibre optical tuneable filter at 2.0 μm is commercial. Please comment on it.
4. A Outlook Section is missing.
5. The quality of the English should be revised carefully.
Comments on the Quality of English Language
The quality of the English should be revised carefully.
Reviewer 4 Report
Comments and Suggestions for Authors
The paper provides an overview of the various different methods used for wavelength tuning in short-pulse and especially mode-locked fiber lasers.
The different methods are divided according to the different physical effect governing the tuning mechanisms and pros and cons for each method are described with a few examples provided.
The topic is interesting however, I find the paper in its current form to be very difficult to read. The lack of any diagrams showing what the author refers to make it necessary to constantly look up the references provided, which is not helpful in a review paper.
At least a few diagrams of the major experimental setups described are essential in my opinion.
I would also highly recommend to have one or more tables collating the information in an easier to digest way.
The author also keeps repeating that some methods are not good (methods that use volume components), however the description of what is good (in-fiber component) is not detailed enough and I would put an emphasis on this part instead.
No significant comments
Round 2
Reviewer 3 Report
Comments and Suggestions for Authors
There is little graphical information. Thus, the paper is very difficult to read.
Comments on the Quality of English LanguageNone.
Author Response
Thank you for your comment. In this connexion, I have added Fig. 2 and inserted more information into Table 1 and into the body of the article. All changes are highlighted in red. The amount of graphic information was augmented and so was the readability of the article. All references were checked and deduplicated.
Round 3
Reviewer 3 Report
Comments and Suggestions for Authors
The responses are mediocre.